# PROMPT-GUIDED DISTILLATION FROM MULTIMODAL LARGE LANGUAGE MODELS TO TASK-SPECIFIC MODELS FOR MULTIMODAL SENTIMENT ANALYSIS

## ABSTRACT

Multimodal Sentiment Analysis (MSA) has made some progress with the advent of Multimodal Large Language Models (MLLMs). However, the scalability and the closed-source nature of some MLLMs imposes challenges for efficient application in the real-word. In this study, we explore an innovative pathway to infuse the capabilities of general MLLMs into task-specific small models for MSA. We introduce the Prompt-Guided Multimodal Framework (PGMF), a refined teacher-student framework designed to transfer knowledge from powerful, general MLLMs to smaller, efficient models. The PGMF-Teacher utilizes MLLM-generated prompts and a tailored conditional alignment module to achieve better MSA, while the PGMF-Student distills this expertise to predict independently of MLLMs' guidance. Extensive evaluations on two popular MSA datasets including SIMS, MOSI and MOSEI demonstrate that compared to previous task-specific small models, PGMF-Teacher achieves state-of-the-art performance with the help of MLLMs, while PGMF-Student achieve competitive results with fewer parameters and without relying on MLLMs' prompts. The proposed framework offers a novel way to equip task-specific small models with the capability of MLLMs.

## 1 INTRODUCTION

Multimodal Sentiment Analysis (MSA) aims to predict sentiment from various types of input, such as language, video, and audio. Accurate MSA is crucial for several downstream applications, such as Human-Computer Interaction and Healthcare (Jiang et al., 2020; Lian et al., 2024). Compared to unimodal sentiment analysis, the mutually complementary nature of multiple modalities typically leads to better performance, thereby improving the applicability of MSA in real-world scenarios.

A series of studies focused on improving MSA through well-designed representational learning and multimodal fusion networks. For example, Tsai et al. (2019a) introduces a novel model, MuLT, which employs multiple Transformers for pairwise alignment of modality information. Hazarika et al. (2020) propose a method to disentangle each modality into modality-invariant and modality-specific features, enabling comprehensive representations of each modality from multiple perspectives for fusion. Additionally, Yu et al. (2021) apply a self-supervised method to generate pseudo-labels for each modality, improving the model's ability to learn modality consistency and variability. Zhang et al. (2023b) make language modality as dominant modality to guide the learning of representations in other modalities, thus mitigating potential conflicts between different modalities. However, after years of research, further performance improvements in small models on MSA datasets have become increasingly challenging.

Meanwhile, multimodal large language models (MLLMs) have demonstrated significantly promise against task-specific small models across various scenarios (Liu et al., 2023; Zhang et al., 2023a; Cheng et al., 2024; Zhao et al., 2024; Wang et al., 2024a). In this context, a recent study (Lian et al., 2024) explores the application of GPT-4V (OpenAI, 2023) for MSA, showing that MLLMs can achieve performance comparable to many task-specific small models. However, the applicability of some MLLMs is limited by their closed-source nature while the applicability of some open-source MLLMs requires large computing resources. These factors limit the application of MLLMs for MSA in real-world scenarios. Additionally, the improvement in accuracy from directly applying the

MLLMs to the MSA task is non-linear with increased parameters, which also limits the real-world application. For example, GPT-4o-mini (OpenAI, 2023) can achieve the F1 of 86.62% on the SIMS dataset, but requires a huge amount of training recources and is only better 4.77% than the current task-specific small SOTA model ALMT (Zhang et al., 2023b).

In this paper, we aims to bridge the gap between small models and MLLMs by leveraging the generalized knowledge from MLLMs to assist in training task-specific small models. To this end, we propose the Prompt-Guided Multimodal Framework (PGMF), which is composed of two parallel streams: PGMF-Teacher and PGMF-Student. In the PGMF-Teacher, a pre-trained MLLMs (*i.e.,* GPT-4o-mini (OpenAI, 2023)) is employed to generate *context-aware prompts* that highlight key sentiment cues across different modalities. These prompts are then used to learn conditional attention maps in designed conditional alignment modules that guide the model to better capture the sentiment information. In the PGMF-Student, we design a similar and smaller model that learns from the guidance provided by the teacher model. It receives the same multimodal inputs but without the prompting of MLLMs. To achieve this, it aligns conditional attention knowledge and related features learned in the teacher model to achieve better MSA tasks while maintaining efficient computation. Extensive experiments on popular datasets, such as SIMS (Yu et al., 2020) and MOSI (Zadeh et al., 2016), validate the effectiveness of PGMF, demonstrating its state-of-the-art performance.

In summary, our work makes the following key contributions: 1) We propose a novel framework that integrates the generalized knowledge of MLLMs to guide smaller, task-specific models for better MSA. The framework leverages a structure composed of two parallel streams, *i.e.,* PGMF-Teacher and PGMF-Student, enabling efficient and effective sentiment prediction across multiple modalities. 2) In the PGMF-Teacher model, we design conditional alignment modules in a simple and straightforward manner to facilitate the prompting of smaller models by large models, thereby enhancing the sentiment analysis capabilities of the teacher models. This design also aids the PGMF-Student model in discarding prompts and achieving efficient MSA independently with few paremeters. 3) Both PGMF-Teacher and PGMF-Student can achieve state-of-the-art performance on several popular datasets (*i.e.,* SIMS, MOSI and MOSEI), especially for PGMF-Student which can achieve improved performance without relying on prompt from MLLMs while maintaining fewer parameters. This approach also offers a novel way to empower task-specific small models with the capabilities of MLLMs.

## 2 RELATED WORK

### 2.1 MULTIMODAL SENTIMENT ANALYSIS

Multimodal Sentiment Analysis (MSA) aims to predict human sentiment by leveraging various types of data, such as video, audio, and text. Early methods, such as Tensor Fusion Networks (TFN) (Zadeh et al., 2017) and Low-rank Multimodal Fusion (LMF) (Liu et al., 2018), achieved state-of-the-art performance by capturing relationships between modalities through Cartesian product-based tensor fusion. However, these methods face the challenge of rapidly increasing computational costs as the feature dimensions and the number of modalities grow. With the advent of deep learning architectures, the attention mechanism has become popular in the design of MSA methods (Tsai et al., 2019a; Rahman et al., 2020; Hazarika et al., 2020; Yuan et al., 2021; Lv et al., 2021; Wang et al., 2023a;b; Zhang et al., 2023b). For example, MulT (Tsai et al., 2019a) employs multi-head attention to align modalities, facilitating more effective multimodal fusion. ALMT (Zhang et al., 2023b) leverages language representations at different scales to guide the learning of auxiliary modalities (*i.e.,* audio and video), effectively mitigating the influence of noise that can negatively impact fusion. In addition, various other novel methods (Han et al., 2021; Yu et al., 2021; Yuan et al., 2024b) have also made significant progress in the MSA. For example, Yu et al. (2021) proposed generating uni-modal sentiment labels to help the model capture both consistency and differentiation across modalities. Moreover, Yuan et al. (2024b) introduced an adversarial training strategy based on semantic reconstruction using original-noisy instance pairs, achieving robust MSA in simulated noisy scenarios.

Despite these advancements, achieving further improvements in performance, especially for small-scale models, remains challenging. A recent study (Lian et al., 2024) explored the application of GPT-4V in MSA, demonstrating that MLLMs can achieve performance comparable to small-

scale models. Different from this work, our work introduces the PGMF framework, which utilizes MLLMs to help the learning of small models rather than directly using MLLMs for MSA.

## 2.2 LARGE LANGUAGE MODELS

In recent years, large language models (LLMs) have made remarkable strides, with models such as GPT-3 (Brown et al., 2020), T5 (Raffel et al., 2020), and LLaMa (Touvron et al., 2023) demonstrating impressive capabilities by scaling both data and model sizes. However, despite these advances, uni-modal LLMs are limited to processing text-based information, restricting their applicability to a broader range of tasks and scenarios. To overcome this limitation, researchers have explored the potential of multimodal large language models (MLLMs), building upon the foundation of uni-modal LLMs. Significant progress has been made in developing powerful MLLMs (Anil et al., 2023; Wang et al., 2023c; Zhu et al., 2024; Maaz et al., 2024; Zhang et al., 2023a; Cheng et al., 2024; Zhao et al., 2024; Li et al., 2023; Dai et al., 2023; Wang et al., 2024b; He et al., 2024), showcasing their surprising practical capabilities. For instance, GPT-4V (OpenAI, 2023) integrates natural language processing with visual understanding to analyze images and provide textual responses to questions about them. Similarly, LLavA (Liu et al., 2023) translates visual content into text by employing a linear layer to embed images, making the LLMs understand visual input. Video-LLaMA (Zhang et al., 2023a) achieving multimodal understanding by aggregating representations from different modalities after applying positional embedding through Q-formers (Li et al., 2023). Moreover, (Zhao et al., 2024) introduced MMICL, which leverages multimodal in-context learning and a specialized dataset to achieve state-of-the-art performance on various visual language tasks. In this work, we utilize GPT-4o-mini, a cost-effective model with a lower token cost, to generate prompts for smaller models, enabling efficient multimodal interactions.

## 2.3 TEACHER-STUDENT MODELS

The teacher-student framework has been widely applied in knowledge distillation, particularly for knowledge compression (Hu et al., 2023). It focuses on transferring knowledge from a larger teacher model to a smaller student model through carefully designed strategies, such as soft label matching (Hinton et al., 2015; Tarvainen & Valpola, 2017; Yuan et al., 2020; 2024a) and feature matching (Romero et al., 2015; Kim et al., 2018; Zagoruyko & Komodakis, 2017; Li et al., 2024). For example, Hinton et al. (2015) introduced the use of the teacher model's probability distribution as soft labels to guide the student model's learning process. By utilizing these soft labels, the student model is trained not only to predict the correct labels but also to closely align with the teacher model's soft predictions, thereby facilitating effective knowledge transfer. Additionally, (Zagoruyko & Komodakis, 2017) proposed an attention transfer method that improves the student model's performance by transferring activation-based and gradient-based attention maps from the teacher model. In the context of MSA, recent advancements include MC-Teacher (Yuan et al., 2024a), which introduced learnable pseudo-label selection and self-adaptive exponential moving average strategies to achieve semi-supervised MSA. In this work, we employ feature matching and attention transfer techniques to achieve our research objectives. To the best of our knowledge, this is the first attempt to transfer the general knowledge of MLLMs to smaller models for MSA.

## 3 METHOD

### 3.1 OVERVIEW

The overall pipeline of the PGMF is illustrated in Figure 1. The framework follows a Teacher-Student model structure, where the PGMF-Teacher is trained independently, and its knowledge is subsequently distilled into the PGMF-Student. First, the PGMF-Teacher Model is trained on preprocessed video, language, and audio input sequences from the datasets. Each modality is processed independently through three embedding layers: Video Embedding, Language Embedding, and Audio Embedding layers. The extracted features from these modalities are then aligned using a designed Conditional Alignment module, where the condition is provided by prompts from MLLMs (*e.g.,* GPT-4o-mini). Specifically, visual and audio features are aligned with language features via two alignment modules: Visual-to-Language (V $\rightarrow$ L) Alignment and Audio-to-Language (A $\rightarrow$ L) Alignment. These conditional alignment layers establish correspondences between modalities

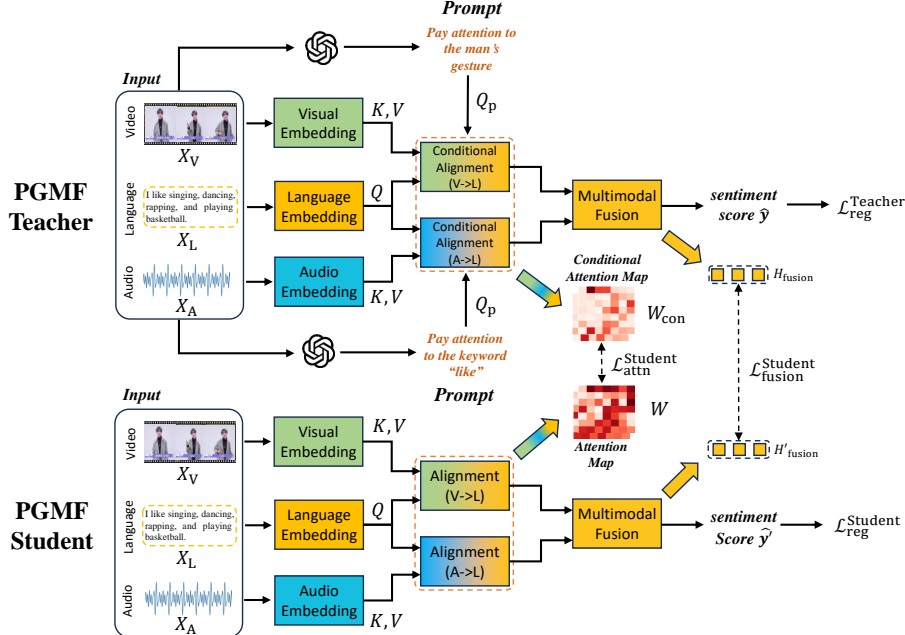

Figure 1: Overall pipeline of PGMF. Note: 1) L, A, and V refer to language, audio, and video/visual, respectively. 2) The language, video, audio inputs are preprocessed as sequences by BERT (Devlin et al., 2019), OpenFace (Baltrusaitis et al., 2018) and Librosa (McFee et al., 2015), respectively. The raw data is displayed for the reader's convenience.

with the help of the prompt, facilitating effective multimodal fusion with the help of MLLMs. The Multimodal Fusion module then combines the aligned features to produce a unified representation, which is used to predict the final sentiment score via a regression loss $L_{\text{reg}}^{\text{Teacher}}$ (defined as Eq. 9).

Once the PGMF-Teacher is trained, a simpler Student model is trained using Knowledge Distillation, where it learns to mimic the behavior of the Teacher model. The key difference between the PGMF-Student and PGMF-Teacher is that the Alignment modules in the PGMF-Student model align video and audio features with language features directly, without the conditional input used in the PGMF-Teacher. Similarly as the PGMF-Teacher, the aligned features are fused through the multimodal fusion to produce a sentiment score. Additionally, instead of using the regression loss of sentiment scores $L_{\text{reg}}^{\text{Student}}$ (defined as Eq. 12), two regularization techniques are used to help the PGMF-Student learn from the PGMF-Teacher: 1) the PGMF-Student's attention maps are trained to match the PGMF-Teacher's conditional attention maps using an attention transfer loss $\mathcal{L}_{\text{attn}}^{\text{Student}}$ (defined as Eq. 10), and 2) the fused unified representations of the PGMF-Student are encouraged to match those of the PGMF-Teacher through a unified representation matching loss $\mathcal{L}_{\text{fusion}}^{\text{Student}}$ (defined as Eq. 11). These loss ensure that the model captures the same underlying patterns as the PGMF-Teacher.

## 3.2 MULTIMODAL INPUT

We utilize the preprocessed sequences from each modality in the datasets as inputs. Specifically, the language input is processed using BERT (Devlin et al., 2019), while visual input is handled by OpenFace (Baltrusaitis et al., 2018), and audio input is processed with Librosa (McFee et al., 2015). We denote the multimodal input as $X_m \in \mathbb{R}^{T_m \times d_m}$, where $m \in \{L, A, V\}$, $T_m$ represents the length of the input sequence, and $d_m$ indicates the vector dimension.

## 3.3 MODALITY EMBEDDING

Given the multimodal input $X_m$, we apply three embedding layers $\text{E}_m$, each consisting of a linear layer to extract features from each modality and map them into a unified feature dimension $d$:

$$S_m = \text{E}_m(X_m, \theta_{\text{E}_m}) \in \mathbb{R}^{T_m \times d}, \tag{1}$$

where $S_m$ represents the embedded features of modality $m$, and $\theta_{\mathrm{E}_m}$ denotes the parameters associated with each embedding layer.

### 3.4 CONDITIONAL ALIGNMENT IN PGMF-TEACHER & ALIGNMENT IN PGMF-STUDENT

In the alignment stage, we aligned the obtained $S_{\mathrm{V}}$ and $S_{\mathrm{A}}$ to $S_{\mathrm{L}}$ using the designed Conditional Alignment module and Alignment module. In PGMF-Teacher, we leverage the condition (*i.e.,* prompts from MLLMs) to help the conditional alignment layers in establishing correspondences between modalities. The MLLMs (*e.g.,* GPT-4o-mini) need to specify which elements in the language and audio inputs require more attention, as well as which visual cues should be emphasized in the visual modality. We denote the aligned outputs of the Conditional Alignment module as $H_{\mathrm{V}\rightarrow\mathrm{L}}^{\mathrm{Teacher}}$ and $H_{\mathrm{A}\rightarrow\mathrm{L}}^{\mathrm{Teacher}}$ which are then utilized for multimodal fusion. For example, the process that align visual modality to language modality can be described as:

$$H_{\mathrm{V}\rightarrow\mathrm{L}}^{\mathrm{Teacher}} = \mathrm{ConditionalAlignment}(X_{\mathrm{V}}, X_{\mathrm{L}} \mid X_{\mathrm{P}}, \theta_{\mathrm{V}\rightarrow\mathrm{L}}^{\mathrm{Teacher}}) \in \mathbb{R}^{T_{\mathrm{L}}\times d}, \tag{2}$$

where $\mathrm{ConditionalAlignment}$ represents the Conditional Alignment module, $X_{\mathrm{P}}$ denotes the prompt from MLLMs, $\theta_{\mathrm{V}\rightarrow\mathrm{L}}^{\mathrm{Teacher}}$ is the parameters used to align the modalities.

In contrast, in PGMF-Student, the designed Alignment module learns the relationships between modalities independently, without prompts from MLLMs. We denote the outputs of the module as $H_{\mathrm{V}\rightarrow\mathrm{L}}^{\mathrm{Student}}$ and $H_{\mathrm{A}\rightarrow\mathrm{L}}^{\mathrm{Student}}$. For example, the $H_{\mathrm{V}\rightarrow\mathrm{L}}^{\mathrm{Student}}$ can be obtained by:

$$H_{\mathrm{V}\rightarrow\mathrm{L}}^{\mathrm{Student}} = \mathrm{Alignment}(X_{\mathrm{V}}, X_{\mathrm{L}}, \theta_{\mathrm{V}\rightarrow\mathrm{L}}^{\mathrm{Student}}) \in \mathbb{R}^{T_{\mathrm{L}}\times d}, \tag{3}$$

where $\mathrm{Alignment}$ and $\theta_{\mathrm{V}\rightarrow\mathrm{L}}^{\mathrm{Student}}$ represent the Alignment module and parameters, respectively.

In the followings, we will further elaborate on each component of the designed Conditional Alignment module and Alignment module: 1) Prompt Embedding, 2) Conditional Alignment in PGMF-Teacher, and 3) Alignment in PGMF-Student. It is important to note that these modules are designed based on the Transformer architecture. For more details on the overall pipeline of the Transformer, we refer readers to Vaswani et al. (2017); Dosovitskiy et al. (2021); Tsai et al. (2019a).

**Prompt Embedding.** To extract features from the MLLMs' prompt $X_{\mathrm{P}}$ and fix the feature dimension to $d$, we apply a pre-trained BERT along with an embedding layer (comprising two layers of Transformer encoders) to $X_{\mathrm{P}}$. We denote the combined operation of the MLLMs, pre-trained BERT, and the embedding layer as $\mathrm{E}_{\mathrm{P}}$. The process can be described as follows:

$$S_{\mathrm{P}} = \mathrm{E}_{\mathrm{P}}(X_{\mathrm{P}}, \theta_{\mathrm{E}_{\mathrm{P}}}) \in \mathbb{R}^{T_{\mathrm{L}}\times d}, \tag{4}$$

where $S_{\mathrm{P}}$ represents the embedded feature of the prompt, which has the same feature shape as $S_{\mathrm{L}}$, and $\theta_{\mathrm{E}_{\mathrm{P}}}$ denotes the parameters used in the MLLMs, pre-trained BERT, and the embedding layer.

**Conditional Alignment in PGMF-Teacher.** The overall architecture of the Conditional Alignment module is similar to the Transformer decoder (Vaswani et al., 2017; Tsai et al., 2019a), with each layer consisting of a our designed conditional attention block and a feed-forward block. In practice, this involves replacing the attention layer in the Transformer decoder with our designed conditional attention layer while keeping the other components unchanged. As illustrated in Figure 2, to align modality $\beta$ to modality $\alpha$, the module first uses $S_{\alpha}$ to compute Query ($Q_{\alpha}$), while $S_{\beta}$ is used to compute the Key ($K_{\beta}$) and Value ($V_{\beta}$). The relationship/attention map $W_{\alpha,\beta}$ between these two modalities is computed as follows:

$$W_{\alpha,\beta} = \frac{Q_{\alpha}K_{\beta}^{\mathrm{T}}}{\sqrt{d_k}} \in \mathbb{R}^{T_{\alpha}\times T_{\beta}}, \tag{5}$$

where $d_k$ denotes the dimension of each attention head, and $T_{\alpha}$ and $T_{\beta}$ represent the sequence lengths of the corresponding modalities. Simultaneously, we apply the prompt $S_{\mathrm{P}}$ as a conditional Query ($Q_{\mathrm{P}}$) to $K_{\beta}$ and $V_{\beta}$ to compute a shifted attention map $\Delta \in \mathbb{R}^{T_{\alpha}\times T_{\beta}}$. Then, we obtained the conditional attention map $W_{con}$ by fusing $W_{\alpha,\beta}$ and $\Delta$:

$$W_{\mathrm{con}} = \mathrm{softmax}(\mathrm{Hadamard}(W_{\alpha,\beta}, \Delta)) \in \mathbb{R}^{T_{\alpha}\times T_{\beta}}, \tag{6}$$


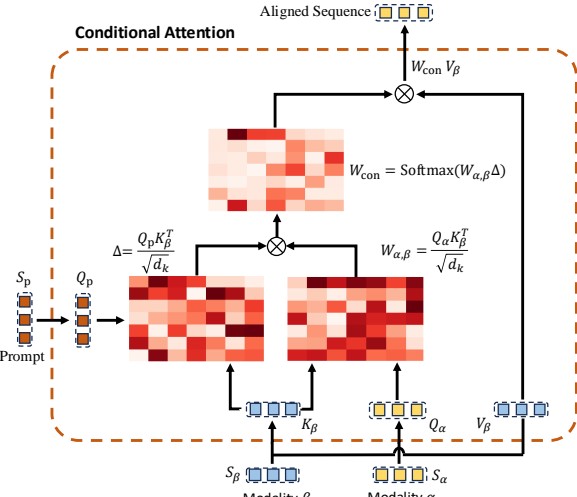

Figure 2: An example of conditional attention used to align modality $\beta$ to modality $\alpha$ under the guidance of a prompt. We denote the aligned sequence after the feed-forward process as $H_{\beta \to \alpha}^{\text{Teacher}}$. Note: 1) $S_{\text{p}}$ represents the prompt features extracted by BERT. 2) All Query, Key, and Value are computed using linear transformations, consistent with the original Transformer architecture.

where the $\mathrm{softmax}$ represents weight normalization operation, $\mathrm{Hadamard}$ represents the Hadamard product operation, which performs an element-wise multiplication of the two attention maps. Finally, the aligned feature $H_{\beta \to \alpha}^{\text{Teacher}}$ can be computed as follows:

$$H_{\beta \to \alpha}^{\text{Teacher}} = \text{Feed-Forward}(W_{\text{con}} V_\beta, \theta_{\text{forward}}) \in \mathbb{R}^{T_\alpha \times d}, \tag{7}$$

where Feed-Forward and $\theta_{\text{forward}}$ represent the MLPs and corresponding parameters. In practice, we utilize two Conditional Alignment modules, each with a depth of six layers, to align the visual and audio modalities to the language modality, respectively. Additionally, similar to the Transformer (Vaswani et al., 2017; Tsai et al., 2019a), we also apply residual connections within the module.

**Alignment in PGMF-Student.** The pipeline of the Alignment is similar to the Conditional Alignment in PGMF-Teacher. The differences is that the PGMF-Student has to independently learn the relationships between modalities without the help of prompts. In practice, we employ two Alignment modules, each with a depth of two layers, to align the visual and audio modilities to the language modality, respectively.

## 3.5 MULTIMODAL FUSION AND PREDICTION

With these features extracted from the various modalities, we employ a Transformer encoder with self-attention blocks for multimodal fusion. In paractice, we concatenate the obtained features with a randomly initialized and learnable regression token $H_{\text{fusion}} \in \mathbb{R}^{1 \times d}$ as input, then the Transformer encoder can transfer and compress essential information to the $H_{\text{fusion}}$, thus making sentiment prediction through this token. For the final sentiment prediction, we apply a linear layer to $H_{\text{fusion}}$:

$$\hat{y} = \text{Regression}(H_{\text{fusion}}, \theta_{\text{regression}}) \in \mathbb{R}^1, \tag{8}$$

where $\hat{y}$ denotes the predicted sentiment score, Regression represents the linear layer, and $\theta_{\text{regression}}$ represents the parameters of the linear layer.

## 3.6 OVERALL LEARNING OBJECTIVES

As outlined in Section 3.1, the training of PGMF consists of two stages: (1) training the PGMF-Teacher and (2) training the PGMF-Student. In the first stage, the PGMF-Teacher learns to perform MSA under the guidance of prompts from MLLMs. The overall learning objective is defined as:

$$\mathcal{L}_{\text{overall}}^{\text{Teacher}} = \mathcal{L}_{\text{reg}}^{\text{Teacher}} = \frac{1}{N} \sum_{i=0}^{N} |\hat{y}^i - y^i|, \tag{9}$$

where $N$ is the number of samples in the training set, $y^i$ is the sentiment label of the $i$-th sample, $\hat{y}^i$ is the prediction of PGMF-Teacher. In the second stage, the PGMF-Student is trained under the supervision of the pre-trained PGMF-Teacher, whose parameters remain frozen. The attention transfer loss $\mathcal{L}_{attn}^{\text{Student}}$ is formulated as follows:

$$\mathcal{L}_{\text{attn}}^{\text{Student}} = \frac{1}{N} \sum_{i=0}^{N} |W^i - W_{\text{con}}^i|, \tag{10}$$

where $W^i$ is the attention map from the last layer of the alignment module in the PGMF-Student, and $W_{\text{con}}^i$ is the conditional attention map from the last layer of the conditional alignment module in PGMF-Teacher. The fused unified representation matching loss $\mathcal{L}_{fusion}^{\text{Student}}$ is defined as:

$$\mathcal{L}_{\text{fusion}}^{\text{Student}} = \frac{1}{N} \sum_{i=0}^{N} |H_{\text{fusion}}'^i - H_{\text{fusion}}^i|, \tag{11}$$

where $H_{\text{fusion}}'^i$ and $H_{\text{fusion}}^i$ represent the fused features from the PGMF-Student and PGMF-Teacher, respectively. The sentiment prediction loss for the PGMF-Student is:

$$\mathcal{L}_{\text{reg}}^{\text{Student}} = \frac{1}{N} \sum_{i=0}^{N} |\hat{y}'^i - y^i|, \tag{12}$$

where $\hat{y}'^i$ is the prediction of PGMF-Student. Overall, the learning objective of PGMF-Student is:

$$\mathcal{L}_{\text{overall}}^{\text{Student}} = \mathcal{L}_{\text{reg}}^{\text{Student}} + \alpha \mathcal{L}_{\text{attn}}^{\text{Student}} + \beta \mathcal{L}_{\text{fusion}}^{\text{Student}}, \tag{13}$$

where the $\alpha$ and $\beta$ are empirically chosen hyperparameters. In practice, for the SIMS dataset, $\alpha$ and $\beta$ are set to 60.0 and 8.0, respectively, while for the MOSI dataset, they are set to 100.0 and 4.0.

## 4 EXPERIMENT AND ANALYSIS

### 4.1 BASELINES

We perform a comprehensive comparison with several advanced methods on MOSI and SIMS datasets, including: TFN (Zadeh et al., 2017), LMF (Liu et al., 2018), MFN (Zadeh et al., 2018), MFM (Tsai et al., 2019b), MuLT (Tsai et al., 2019a), MISA (Hazarika et al., 2020), Self-MM (Yu et al., 2021), TETFN (Wang et al., 2023a) and ALMT (Zhang et al., 2023b).

### 4.2 EVALUATION CRITERIA

Consistent with previous works (Hazarika et al., 2020; Zhang et al., 2023b), we evaluate the regression tasks by reporting the mean absolute error (MAE) and the correlation between the model's predictions and human annotations (Corr). Additionally, sentiment predictions can be classified as either negative/positive or negative/non-negative based on the sentiment score. We also report binary classification accuracy (Acc-2) and the weighted F1-score (F1) on both datasets. Specifically, Acc-2 and F1 are reported based on negative/non-negative classification for both datasets. To make a comprehensive comparison with previous methods, we also report Acc-2 and F1 scores based on negative/positive classification for the MOSI dataset. In the tables, performance metrics computed using these two classification methods are separated by a "/", with the left side representing negative/non-negative performance and the right side representing negative/positive performance. All results are averaged over 5 runs and standard deviations are reported. In addition, we focus on comparing the designed components. Therefore, parameters from BERT used for input preprocessing in all models are excluded from the reported parameter count for comparison purposes.

### 4.3 COMPARISON RESULTS

Table 1, Table 2 and Table 3 present the comparative results on the SIMS, MOSI and MOSEI datasets, respectively. Notably, the performance of the PGMF-Teacher is close to the MLLMs (*e.g.,* GPT-4o-mini) in many metrics, and it outperforms both Video-LLaMA2 and GPT-4V in all metrics

Table 1: Performance Comparison on SIMS dataset. Note: 1) $a$ represents the results reproduced by the authors from open-source code with default hyperparameters. 2) $b$ represents the results are from Lian et al. (2024). 3) $c$ represents the resluts are from Yu et al. (2020).

| Method | Parm. | Acc-2 (↑) | F1 (↑) | MAE (↓) | Corr (↑) |
|---|---|---|---|---|---|
| Video-LLaMA2[a] | 7B | 80.09 | 79.94 | 0.584 | 0.476 |
| GPT-4V[b] | - | 81.24 | - | - | - |
| GPT-4o-mini[a] | - | **86.48** | **86.62** | **0.453** | **0.663** |
| MFN[c] | - | 77.86±0.4 | 78.22±0.4 | 0.452±1.2 | 0.552±0.2 |
| MuLT[c] | - | 77.94±0.9 | 79.10±0.9 | 0.485±2.6 | 0.559±0.6 |
| TFN[c] | - | 80.66±1.4 | 81.62±1.1 | 0.425±1.1 | 0.612±1.2 |
| LMF[c] | - | 79.34±0.4 | 79.96±0.6 | 0.440±1.6 | 0.600±1.3 |
| TFN[a] | 35.63M | 78.12±1.56 | 77.83±1.62 | 0.434±1.12 | 0.579±1.50 |
| MISA[a] | 21.66M | 77.72±1.10 | 76.54±1.67 | 0.451±1.83 | 0.570±1.95 |
| Self-MM[a] | 0.38M | 77.94±1.11 | 77.72±0.68 | 0.418±1.05 | 0.589±1.54 |
| TETFN[a] | 1.53M | 80.18±0.49 | 79.34±0.52 | 0.422±1.30 | 0.588±1.71 |
| ALMT[a] | 2.60M | 79.91±0.29 | 80.17±0.60 | 0.421±0.69 | 0.583±0.70 |
| **PGMF** | | | | | |
| *Teacher* | 2.54M | **83.06±0.95** | **84.06±0.43** | **0.370±0.50** | **0.690±0.80** |
| *Student* | 0.82M | 81.40±1.58 | 81.85±1.41 | 0.382±1.39 | 0.662±1.26 |

Table 2: Performance Comparison on MOSI dataset. Note: 1) $a$ represents the results reproduced by the authors from open-source code with default hyperparameters. 2) $b$ represents the results are from Lian et al. (2024).

| Method | Parm. | Acc-2 (↑) | F1 (↑) | MAE (↓) | Corr (↑) |
|---|---|---|---|---|---|
| Video-LLaMA2[a] | 7B | 83.24/86.43 | 82.60/86.23 | 1.149 | 0.696 |
| GPT-4V[b] | - | 80.43/- | - | - | - |
| GPT-4o-mini[a] | - | **87.32/89.48** | **87.17/89.42** | **0.997** | **0.842** |
| TFN[a] | 9.50M | 77.38±1.37/78.11±0.60 | 77.35±1.33/78.02±0.57 | 0.949±3.13 | 0.662±1.95 |
| MISA[a] | 1.14M | 80.93±0.99/81.05±0.83 | 80.90±1.03/81.01±0.87 | 0.773±1.81 | 0.775±0.63 |
| Self-MM[a] | 0.16M | 82.94±0.63/83.18±0.35 | 82.95±0.63/83.09±0.36 | 0.717±1.53 | 0.792±0.55 |
| TETFN[a] | 1.25M | 80.87±0.52/80.82±0.53 | 80.87±0.52/80.82±0.53 | 0.726±1.68 | 0.791±0.86 |
| ALMT[a] | 2.50M | 83.00±0.22/85.12±0.20 | 83.00±0.22/85.19±0.27 | **0.713±0.75** | 0.795±0.54 |
| **PGMF** | | | | | |
| *Teacher* | 1.45M | **85.05±0.66/86.61±0.69** | **85.15±0.66/86.69±0.69** | 0.734±1.46 | **0.797±0.60** |
| *Student* | 0.53M | 83.62±0.91/85.37±1.00 | 83.68±0.96/85.50±0.96 | 0.746±1.63 | 0.775±1.10 |

on both datasets. Interestingly, PGMF-Teacher surpasses all MLLMs on MAE and Corr. For example, on the SIMS, the PGMF-Teacher achieves a MAE of 0.370±0.50, outperforming GPT-4o-mini (0.453). This indicates that task-specific models may outperform general-purpose MLLMs using zero-shot prompting in certain scenarios. Furthermore, compared to Video-LLaMA2 and GPT-4V, both the PGMF-Teacher and PGMF-Student demonstrate improvements across most metrics. For example, on the SIMS, the PGMF-Student achieves an Acc-2 of 81.40±1.58, marking a relative improvement of 1.64% over Video-LLaMA2. When compared to the task-specific small model ALMT, PGMF-Student achieves a 2.10% relative improvement in F1 on the SIMS. A similar trend is observed on the MOSI dataset (Table 2), showing the general applicability of PGMF across cultures, *i.e.,* both Chinese and English datasets. Moreover, it is worth noting that the PGMF-Student can achieve advanced performance with fewer parameters compared to MLLMs, which underscores the potential of task-specific small models in the MSA field. This demonstrates that smaller models are not necessarily inferior to general larger models in all situations. Furthermore, as shown in the Table 3, the results on the larger dataset (MOSEI) show that PGMF-Teacher/-Student achieves advanced performance on most of the metrics with few parameters. This demonstrates that PGMF has good generalization ability on data sets of different sizes. It is worth noting that Self-MM with the fewest parameters shows well performance on the MOSEI dataset. This also demonstrates that the feasibility of suitable strategies to achieve strong performance with smaller parameters.

Table 3: Performance Comparison on MOSEI dataset. Note: $a$ represents the results reproduced by the authors from open-source code with default hyperparameters.

| Method | Parm. | Acc-2 (↑) | F1 (↑) | MAE (↓) | Corr (↑) |
|---|---|---|---|---|---|
| Video-LLaMA2[a] | 7B | 83.29/84.50 | 83.23/85.21 | **0.922** | 0.406 |
| GPT-4o-mini[a] | - | **85.04/86.90** | **85.25/87.04** | 1.015 | **0.744** |
| TFN[a] | 5.04M | 83.00±0.45/82.90±0.43 | 82.68±0.40/82.83±0.41 | 0.566±0.31 | 0.725±0.21 |
| MISA[a] | 1.14M | 84.41±0.30/85.09±0.62 | 84.16±0.30/85.02±0.59 | 0.553±0.46 | 0.759±0.25 |
| Self-MM[a] | 0.16M | 84.15±0.50/84.90±0.49 | 84.15±0.43/84.79±0.40 | **0.529±0.47** | 0.764±0.45 |
| TETFN[a] | 1.25M | 84.18±0.62/85.42±0.43 | 84.06±0.63/85.31±0.55 | 0.543±0.51 | 0.769±0.27 |
| ALMT[a] | 3.21M | 84.35±0.34/84.76±0.45 | 84.10±0.32/84.25±0.59 | 0.542±0.45 | 0.768±0.17 |
| **PGMF** | | | | | |
| *Teacher* | 1.47M | **85.08±0.36/86.62±0.75** | **85.55±0.24/86.71±0.71** | 0.539±1.06 | **0.773±1.51** |
| *Student* | 0.48M | 83.96±0.38/84.67±0.27 | 84.20±0.48/84.74±0.28 | 0.548±0.41 | 0.747±0.51 |

## 4.4 EFFECT OF EACH COMPONENT

To evaluate the impact of each component, we conducted experiments by removing specific components. First, when we removed the MLLMs' prompt from the PGMF-Teacher, we observed a significant drop in performance across both datasets. Specifically, on the SIMS dataset, the F1 score decreased from 84.06% to 80.84%, and MAE increased from 0.370 to 0.436. A similar trend was observed on the MOSI dataset, where the F1 score dropped from 85.15% to 79.60%, and MAE increased from 0.734 to 0.914. These phenomenoa show that the MLLMs plays a crucial role in helping the model capture relevant multimodal information more effectively. Second, we removed the guidance of the PGMF-Teacher during the training of the PGMF-Student. This led to a decrease in the student model's performance, with the F1 score on SIMS dropping from 81.85% to 78.72%, and on MOSI from 83.68% to 83.00%. The increase in MAE values on both datasets also reflects the PGMF-Student model's reduced ability to align multimodal information without teacher guidance. It also shows that the importance of knowledge distillation, as the PGMF-Teacher's guidance can help the PGMF-Student learn the relationship between each modality effectively.

In addition, we also observed that the guidance from the PGMF-Teacher had a greater impact on the student model's performance on the SIMS dataset compared to the MOSI dataset. We believe that this difference may be because of the diversity of data in the SIMS dataset. Specifically, the data of SIMS dataset contains complex environments and disturbances such as lighting, head pose and audio background noise. This makes the data difficult for the PGMF-Student to achieve better performance without relying on the guidance of the PGMF-Teacher.

Table 4: Effect of Each Component.

| Method | SIMS | | MOSI | |
|---|---|---|---|---|
| | F1 | MAE | F1 | MAE |
| **PGMF-Teacher** | **84.06±0.43** | **0.370±0.50** | **85.15±0.66/86.69±0.69** | **0.734±1.46** |
| *w/o prompt* | 80.84±0.93 | 0.436±0.57 | 79.60±0.95/81.21±1.07 | 0.914±0.68 |
| **PGMF-Student** | **81.85±1.41** | **0.382±1.39** | **83.68±0.96/85.50±0.96** | **0.746±1.63** |
| *w/o guidance of teacher* | 78.72±0.53 | 0.429±1.02 | 83.00±0.59/85.07±0.52 | 0.743±1.30 |

## 4.5 EFFECT OF EACH REGULARIZATION

To evaluate the effect of each regularization in the PGMF-Student, we removed $\mathcal{L}_{\text{attn}}^{\text{Student}}$, $\mathcal{L}_{\text{fusion}}^{\text{Student}}$, and both $\mathcal{L}_{\text{fusion}}^{\text{Student}}$ and $\mathcal{L}_{\text{attn}}^{\text{Student}}$, and observed the model's performance on the SIMS and MOSI datasets. The results are presented in Table 5. We observe that both F1 and MAE decrease when each regularization is removed, indicating that every regularization contributes positively to the performance of PGMF-Student. Moreover, it is evident that the impact of each regularization is more significant on the SIMS dataset than on the MOSI dataset. For example, when $\mathcal{L}_{\text{attn}}^{\text{Student}}$ is removed, the F1 score drops by a relative 3.24% on SIMS, while it decreases by only 1.11% on MOSI. These differ-

ences could be attributed to the varying levels of difficulty between the SIMS and MOSI datasets. Additionally, we tried different combinations of $\alpha$ and $\beta$, please see Appendix C.2 for more details.

Table 5: Effect of Each Regularization.

| Method | SIMS | | MOSI | |
| --- | --- | --- | --- | --- |
| | F1 | MAE | F1 | MAE |
| **PGMF-Student** | **81.85±1.41** | **0.382±1.39** | **83.68±0.96/85.50±0.96** | **0.746±1.63** |
| w/o $\mathcal{L}_{attn}^{Student}$ | 79.28±0.75 | 0.453±0.48 | 82.76±0.30/84.80±0.42 | 0.741±0.71 |
| w/o $\mathcal{L}_{fusion}^{Student}$ | 79.23±0.69 | 0.428±0.87 | 83.16±0.51/85.44±0.55 | 0.738±0.76 |
| w/o $\mathcal{L}_{fusion}^{Student}$ & $\mathcal{L}_{attn}^{Student}$ | 78.72±0.53 | 0.429±1.02 | 83.00±0.59/85.07±0.52 | 0.743±1.30 |

## 4.6 EFFECT OF MLLMS' PROMPTS

To intuitively verify the effect of MLLMs' prompts, we first collected the conditional attention map and the attention map without MLLMs' prompts from the PGMF-Teacher. As shown in Figure 3, we visualized the attention difference maps by subtracting the attention map without MLLMs' prompts from the conditional attention map $H_{V \to L}^{Teacher}$. Obviously, with MLLMs' prompts, the model is able to focus more on key words in the language and key frames in the video, demonstrating that the PGMF-Teacher benefits significantly from the guidance of MLLMs' prompts. This improvement also lays the foundation for the PGMF-Student to achieve better performance in MSA.

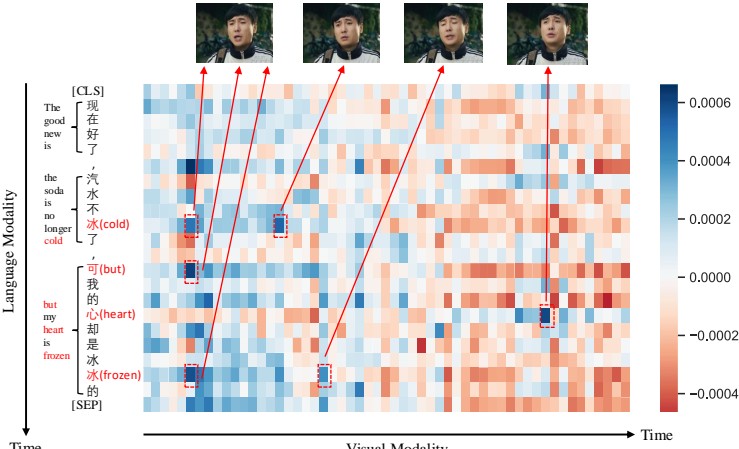

Figure 3: An example of attention difference maps on the SIMS. This difference map is obtained by subtracting the attention map without MLLMs' prompts from the conditional attention map $H_{V \to L}^{Teacher}$. Note: The blue areas indicate regions where the model focuses more when guided by the prompts, while the orange areas indicate regions where the model focuses less under the same prompts.

## 5 CONCLUSION

In this paper, we propose a novel Prompt-Guided Multimodal Framework (PGMF) to enhance Multimodal Sentiment Analysis (MSA) with the assistance of general MLLMs (*e.g.,* GPT-4o-mini). The framework is built on a teacher-student architecture, where the MLLMs' prompt serves as a conditional input to guide the learning of the PGMF-Teacher. This knowledge is then further distilled into the PGMF-Student, allowing it to learn independently without the support of MLLMs. Extensive comparative experiments and ablation studies demonstrate the effectiveness of PGMF, providing new insights into utilizing MLLMs for improved MSA.

## ETHICS STATEMENT

All experiments in this study are conducted using publicly available datasets. We have reported our findings in an objective and responsible manner. Therefore, we believe that this work does not pose ethical issues.

## REPRODUCIBILITY STATEMENT

We have made several efforts to ensure the reproducibility of our results. The details required to reproduce the PGMF can be found in Section 3, Section 4 and Appendix B. In addition, we will make the code publicly available to facilitate the reproduction of our results after the paper is accepted. We encourage the community to reproduce our results using the released code and to refer to the results based on the five runs average with different random seeds for a comprehensive comparison.

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

## A  DATASETS

We conducted extensive experiments on two popular MSA datasets *i.e.,* SIMS (Yu et al., 2020) and MOSI (Zadeh et al., 2016).

### A.1  SIMS

SIMS is a Chinese MSA dataset, with data sourced from Chinese movies, TV series, and variety shows, featuring complex real-world scenarios. It consists of 1,368 training samples, 456 validation samples, and 457 test samples. Each sample is manually annotated with a continuous sentiment score ranging from -1 to 1, where -1 represents highly negative sentiment, and 1 represents highly positive sentiment.

### A.2  MOSI

MOSI is an English MSA dataset, composed of data collected from YouTube. The dataset includes 1,284 training samples, 229 validation samples, and 686 test samples. Each instance is manually annotated with a continuous sentiment score ranging from -3 to 3, with -3 representing highly negative sentiment and 3 representing highly positive sentiment, similar to SIMS.

### A.3  MOSEI

MOSEI is an English MSA dataset with data collected from YouTube. It contains 22,856 video clips, including 16,326 training samples, 1,871 validation samples, and 4,659 test samples. Similar to MOSI, each sample is manually annotated with a score ranging from -3 to 3.

## B  IMPLEMENTATION DETAILS

### B.1  HYPERPARAMETERS

We implemented our proposed method using PyTorch 2.1.1 with CUDA 12.1. The experiments were conducted on a PC equipped with an AMD EPYC 7513 processor (2.6GHz) and an NVIDIA Tesla A40 GPU. The key parameters are listed in Table 6.

In the training of the PGMF-Teacher, we perform random mask on the multimodal input to improve the data diversity. The ratio of random masks is between 0 and 70% on the SIMS dataset and between 0 and 50% on the MOSI and MOSEI datasets. Additionally, since GPT-4o-mini does not support speech analysis, we prompted it to infer possible speech cues based on the available language information. The prompt template used for this task is shown in Listing B.1.

### B.2  PROMPTING TEMPLATE TO GENERATE PROMPTS FOR PGMF-TEACHER

Listing B.1 provides the prompting template used to generate prompts for the PGMF-Teacher on the MOSI dataset. Since SIMS is a Chinese dataset, we directly translated this template into Chinese to generate prompts for the PGMF-Teacher on the SIMS dataset. We can see that there is a strong guidance for prediction in the hints given by the MLLMs. Based on these prompts, the PGMF-Teacher is more easily learn the alignment between modalities and in turn transfer this knowledge to the PGMF-Student which does not rely on MLLMs' prompts. More examples can be seen in Appendix C.5.

Table 6: The parameters used on the SIMS, MOSI and MOSEI datasets

| Parameter | SIMS | MOSI | MOSEI |
|---|---|---|---|
| Common | | | |
| Batch Size | 64 | 64 | 64 |
| Optimizer | AdamW | AdamW | AdamW |
| Epochs | 200 | 200 | 200 |
| Seeds | 1111-1115 | 1111-1115 | 1111-1115 |
| Warm Up | ✓ | ✓ | ✓ |
| Cosine Annealing | ✓ | ✓ | ✓ |
| $d$ | 64 | 64 | 64 |
| $T_{\mathrm{L}}, T_{\mathrm{V}}, T_{\mathrm{A}}, T_{\mathrm{P}}$ | 50, 55, 400, 50 | 50, 500, 375, 50 | 50, 500, 500, 50 |
| The Depth of Language Embedding | 1 | 1 | 1 |
| The Depth of Visual Embedding | 1 | 1 | 1 |
| The Depth of Audio Embedding | 1 | 1 | 1 |
| The Depth of Prompt Embedding | 2 | 2 | 2 |
| MLLMs (GPT-4o-mini) | | | |
| Temperature | 0 | 0 | 0 |
| Version | 2024-07-18 | 2024-07-18 | 2024-07-18 |
| PGMF-Teacher | | | |
| Initial Learning Rate | 1e-4 | 1e-4 | 2e-4 |
| The Depth of Conditional Alignment | 6 | 6 | 6 |
| The Depth of Multimodal Fusion | 6 | 6 | 6 |
| PGMF-Student | | | |
| $\alpha, \beta$ | 60.0, 8.0 | 100.0, 4.0 | 100.0, 4.0 |
| Initial Learning Rate | 2e-4 | 1e-4 | 2e-4 |
| The Depth of Conditional Alignment | 2 | 2 | 1 |
| The Depth of Multimodal Fusion | 2 | 2 | 2 |

**Listing B.1: Prompting Template of MLLMs to Generate Prompts for PGMF-Teacher**

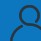 User

### Background ###
You are a multimodal sentiment analysis expert. We provide you with a video-text pair and corresponding sentiment label. Please provide detailed hints to help task-specific small models identify sentiment cues.

### Video-text Pair and Corresponding Label ###
**Note: The label is None if the data is from the test set.**
Video: Please refer to the video input.
Text: THAT HUH BEFORE WE GO ON WITH THAT I MUST SAY THAT THIS HAD A SURPRISINGLY DECENT CAST
Label: Positive

### Response Requirements ###
1. Focus on facial expressions in the video, especially smiles, frowns, and eye movements.
2. Focus on the language in the video, especially emotionally charged words and phrases.

### Output Format ###
Your output consists of the following parts:
1. Visual cues: Observations related to facial expressions in the video.
2. Language cues: Important sentiment cues found in the provided text.
3. Multimodal cues: Combined insights from both video and text that highlight key sentiment cues.

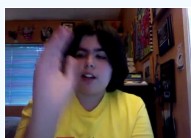 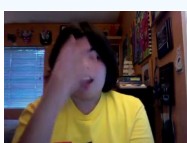 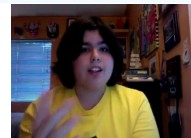

GPT-4o-mini

### Visual Cues ###
1. Facial Expressions: - Look for any smiles or positive expressions that indicate enjoyment or enthusiasm. A genuine smile can enhance the positive sentiment.
- Observe the eyes; wide eyes can convey excitement or surprise, while squinting might indicate skepticism.
2. Body Language: - Notice the gestures; open hand movements can suggest openness and positivity. If the individual is using their hands expressively, it may indicate engagement with the topic.
- Pay attention to posture; an upright and relaxed posture can signal confidence and positivity, while slouching might suggest disinterest.

### Language Cues ###
1. Emotionally Charged Words: - The phrase "surprisingly decent cast" carries a positive connotation. The word "surprisingly" suggests an unexpected positive outcome, which can amplify the sentiment.
- The use of "decent" indicates a favorable opinion, suggesting that the speaker is pleasantly surprised by the quality.

### Multimodal Cues ###
1. Combined Insights: - If the individual displays a smile or enthusiastic gestures while stating "surprisingly decent cast," it reinforces the positive sentiment expressed in the text.
- The alignment of positive facial expressions and body language with the positive language cues creates a strong multimodal signal of approval and enjoyment regarding the cast.

# C  ADDITIONAL EXPERIMENTS

## C.1  GENERALITY OF THE PROPOSED FRAMEWORK

To evaluate the generality of the PGMF and provide a comprehensive comparison with state-of-the-art methods, we applied the Teacher-Student framework to ALMT (Zhang et al., 2023b). As shown in Table 7, ALMT-Teacher outperformed PGMF-Teacher across all metrics on both the SIMS and MOSI datasets, demonstrating the effectiveness of utilizing MLLMs to improve the learning of task-specific small models. However, ALMT-Student did not exhibit the same level of improvement as PGMF-Student. We attribute this isbecause that ALMT was not originally designed with the Teacher-Student framework. Its reliance on multiple specialized attention maps complicates the optimization of the student model during the knowledge distillation process. Additionally, it is worth noting that PGMF-Student achieved better results than ALMT-Student with a significantly smaller number of parameters, further demonstrating the effectiveness and efficiency of the PGMF.

Table 7: Generality of the proposed framework. Note: 1) the best result of each segment is highlighted in bold. 2) $a$ represents the results reproduced by the authors from open-source code with default hyperparameters. 3) $b$ represents the results are from Lian et al. (2024).

| Method | Parm. | Acc-2 ($\uparrow$) | F1 ($\uparrow$) | MAE ($\downarrow$) | Corr ($\uparrow$) |
|---|---|---|---|---|---|
| | | SIMS | | | |
| ALMT[a] | 2.60M | 79.91±0.29 | 80.17±0.60 | 0.421±0.69 | 0.583±0.70 |
| **ALMT w/ Prompt** | | | | | |
| *Teacher* | 2.60M | **84.20±0.57** | **84.45±0.81** | **0.363±0.76** | **0.711±1.50** |
| *Student* | 2.60M | 79.87±1.81 | 80.58±1.05 | 0.418±2.15 | 0.587±3.97 |
| PGMF w/o Prompt | 0.82M | 73.74±4.54 | 80.84±0.93 | 0.436±0.57 | 0.569±0.86 |
| **PGMF** | | | | | |
| *Teacher* | 2.54M | **83.06±0.95** | **84.06±0.43** | **0.370±0.50** | **0.690±0.80** |
| *Student* | 0.82M | 81.40±1.58 | 81.85±1.41 | 0.382±1.39 | 0.662±1.26 |
| | | MOSI | | | |
| ALMT[a] | 2.50M | 83.00±0.22/85.12±0.20 | 83.00±0.22/85.19±0.27 | 0.713±0.75 | 0.795±0.54 |
| **ALMT w/ Prompt** | | | | | |
| *Teacher* | 2.50M | **86.56±0.68/88.02±0.67** | **86.63±0.69/88.06±0.68** | **0.677±0.57** | **0.834±0.46** |
| *Student* | 2.50M | 83.26±0.41/85.43±0.14 | 83.38±0.31/85.52±0.15 | 0.720±0.54 | 0.784±0.28 |
| PGMF w/o Prompt | 0.53M | 79.33±0.79/80.92±0.94 | 79.60±0.95/81.21±1.07 | 0.914±0.68 | 0.675±0.32 |
| **PGMF** | | | | | |
| *Teacher* | 1.45M | **85.05±0.66/86.61±0.69** | **85.15±0.66/86.69±0.69** | **0.734±1.46** | **0.797±0.60** |
| *Student* | 0.53M | 83.62±0.91/85.37±1.00 | 83.68±0.96/85.50±0.96 | 0.746±1.63 | 0.775±1.10 |

## C.2  EFFECT OF REGULARIZATION WEIGHT ON MODEL PERFORMANCE

To investigate the impact of regularization weights, we experimented with various combinations of $\alpha$ and $\beta$ on the SIMS dataset. The results are presented in Table 8. It is evident that both $\alpha$ and $\beta$ influence the performance of the PGMF-Student.

## C.3  PERFORMANCE IMPACT OF VARYING PGMF-STUDENT PARAMETERS

Table 9 presents the performance impact of different parameter settings on the PGMF-Student model. Notably, the PGMF-Student achieves optimal performance with 0.82M parameters, corresponding to a configuration (as shown in Table 6) of 1 embedding layers, 2 alignment layers, and 2 multimodal fusion layers. beyond this point, increasing the model size does not significantly improve the performance, suggesting that the model has likely already fully utilized its learning capacity.

Table 8: Effect of regularization weight on model performance

| $\alpha$ | $\beta$ | Acc-2 ($\uparrow$) | F1 ($\uparrow$) | MAE ($\downarrow$) | Corr ($\uparrow$) |
|---|---|---|---|---|---|
| 60.0 | 8.0 | **81.40$\pm$1.58** | **81.85$\pm$1.41** | **0.382$\pm$1.39** | **0.662$\pm$1.26** |
| 80.0 | 8.0 | 81.01$\pm$1.51 | 81.27$\pm$1.34 | 0.394$\pm$1.40 | 0.650$\pm$2.36 |
| 40.0 | 8.0 | 81.18$\pm$1.66 | 81.44$\pm$1.52 | 0.388$\pm$1.15 | **0.662$\pm$1.58** |
| 20.0 | 8.0 | 80.79$\pm$1.29 | 81.46$\pm$1.17 | 0.387$\pm$1.33 | 0.661$\pm$1.53 |
| 0 | 8.0 | 77.94$\pm$1.12 | 79.28$\pm$0.75 | 0.453$\pm$0.48 | 0.524$\pm$1.87 |
| 60.0 | 10.0 | 81.01$\pm$1.87 | 81.27$\pm$1.67 | 0.389$\pm$1.20 | 0.656$\pm$1.30 |
| 60.0 | 6.0 | 80.88$\pm$1.26 | 81.37$\pm$0.92 | 0.392$\pm$1.53 | 0.653$\pm$1.82 |
| 60.0 | 4.0 | 80.74$\pm$1.01 | 81.23$\pm$1.16 | 0.393$\pm$1.63 | 0.650$\pm$2.26 |
| 60.0 | 2.0 | 80.53$\pm$0.97 | 81.05$\pm$0.99 | 0.396$\pm$1.06 | 0.645$\pm$2.29 |
| 60.0 | 0 | 78.29$\pm$0.42 | 79.23$\pm$0.69 | 0.428$\pm$0.87 | 0.564$\pm$3.10 |
| 0 | 0 | 78.56$\pm$0.44 | 78.72$\pm$0.53 | 0.429$\pm$1.02 | 0.567$\pm$1.39 |

Table 9: Performance Comparison of Varying Student Model Parameters on SIMS dataset. Note: Parameters from BERT used for input preprocessing in all models are excluded from the reported parameter count for fair comparison.

| Method | Parm. | Acc-2 ($\uparrow$) | F1 ($\uparrow$) | MAE ($\downarrow$) | Corr ($\uparrow$) |
|---|---|---|---|---|---|
| ALMT[a] | 2.60M | 79.91$\pm$0.29 | 80.17$\pm$0.60 | 0.421$\pm$0.69 | 0.583$\pm$0.70 |
| PGMF-Student | 0.49M | 80.74$\pm$1.16 | 81.44$\pm$1.03 | 0.408$\pm$1.52 | 0.638$\pm$2.15 |
| PGMF-Student | 0.82M | **81.40$\pm$1.58** | 81.85$\pm$1.41 | **0.382$\pm$1.39** | **0.662$\pm$1.26** |
| PGMF-Student | 1.46M | 80.66$\pm$0.51 | 81.47$\pm$0.54 | 0.400$\pm$1.64 | 0.631$\pm$1.72 |
| PGMF-Student | 2.11M | 81.36$\pm$1.29 | **82.32$\pm$0.75** | 0.394$\pm$1.33 | 0.646$\pm$1.43 |
| PGMF-Student | 4.05M | **81.40$\pm$0.71** | 81.79$\pm$0.50 | 0.394$\pm$1.65 | 0.636$\pm$1.93 |

## C.4 VISUALIZATION OF CONVERGENCE PERFORMANCE

In Figure 4, we visualize the loss curves of PGMF-Student on the SIMS and MOSI datasets. While the overall trend shows a decrease, the variance of $\mathcal{L}_{\text{attn}}^{\text{Student}}$ across different seeds is relatively high. We believe this is due to the difficulty PGMF-Student faces in aligning with the PGMF-Teacher's learning outcomes without the help of MLLMs' prompts, resulting in fluctuations during the optimization process. Despite this, PGMF-Student still achieves SOTA performance on both the SIMS and MOSI datasets, demonstrating the effectiveness of the proposed PGMF framework.

## C.5 EXAMPLES OF PROMPTS

As shown in Figure 5, we provide more examples of MLLMs' prompts, both in Chinese and English. For efficiency and cost-effectiveness, we uniformly sample three frames from the video input as the input to the MLLMs, consistent with previous works (Lian et al., 2024). Since GPT-4o-mini does not support speech data analysis, we did not include speech input, and instead expect the model to infer the corresponding cues from the language input.

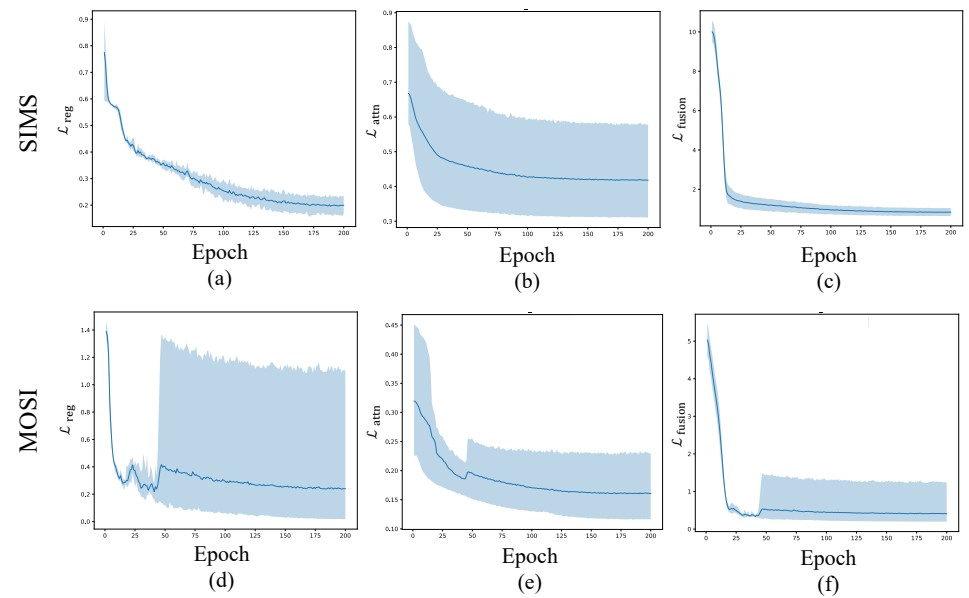

Figure 4: Visualization of convergence performance. Note: The values on the curve represent the averages corresponding to five different random seeds.

**Video**

**Language**

BUT I THINK THERE WAS ALSO SOME SORT OF LIKE UM TONE WITHIN THE ACTUAL UM POWERS OF THE GREEN LANTERN AND HIMSELF AND THAT YOU CAN BASICALLY THINK UP OF ANYTHING AND YOU CAN PROJECT THAT INTO THE REAL LIFE

**Generated Prompts**

### Visual Cues ###
1. **Facial Expressions**:
  - Look for any smiles or positive expressions that indicate enthusiasm or excitement about the topic.
  - Notice if the speaker's eyes widen or if there are any brightening expressions that suggest engagement with the subject matter.

2. **Body Language**:
  - Observe gestures such as hand movements that may emphasize points being made, indicating passion or interest.
  - Pay attention to posture; an open and relaxed posture can suggest confidence and positivity, while a closed posture might indicate uncertainty.

### Language Cues ###
1. **Emotionally Charged Words**:
  - The phrase "think up of anything" suggests creativity and imagination, which are often associated with positive feelings.
  - The use of "project that into real life" implies a sense of empowerment and possibility, reinforcing a positive sentiment.

2. **Tone and Delivery**:
  - Listen for an upbeat tone or inflection in the speaker's voice that conveys excitement or enthusiasm about the topic.

### Multimodal Cues ###
1. **Combined Insights**:
  - The speaker's positive facial expressions, such as smiles or bright eyes, align with the optimistic language in the text, creating a cohesive positive sentiment.
  - Gestures that accompany phrases like "think up of anything" can enhance the feeling of creativity and possibility, further supporting the positive sentiment conveyed in the text.

(a) Example 1

**Video**

**Language**

他是五花大绑，把你全部都绑起来
(He has tied you up tightly, fully restrained)

**Generated Prompts**

### 视觉线索(Visual Cues) ###
1. **面部表情(Facial Expressions)**:
  - **微笑(Smiling)**: 观察是否有微笑的出现，微笑通常传达积极情感(Observe whether a smile appears, as it usually conveys positive emotions)。
  - **皱眉(Frowning)**: 注意是否有皱眉的情况，这可能表示困惑或不满(Pay attention to any frowning, which may indicate confusion or dissatisfaction)。
  - **眼神(Eye Contact)**: 眼神的变化（如注视、回避）可以反映出情感的强度和态度(Changes in eye contact, such as staring or avoiding eye contact, can reflect the intensity and attitude of emotions)。

2. **肢体语言(Body Language)**:
  - **手势(Gestures)**: 手势的使用频率和类型（如指向、握拳）可以传达情感的强烈程度(The frequency and type of gestures, such as pointing or clenching fists, can convey the intensity of emotions)。
  - **姿态(Postures)**: 身体的姿态（如前倾、后仰）可能表明对话题的兴趣或抵触(The posture of the body, such as leaning forward or reclining, may indicate interest in or resistance to the topic being discussed)。

### 语言线索(Language Cues) ###
1. **情感词汇(Emotionally Charged Words)**:
  - "五花大绑" 暗示一种束缚感，可能传达出无助或压迫的情感(The phrase "tied up tightly" suggests a sense of being bound, potentially conveying feelings of helplessness or oppression)。
  - "把你全部都绑起来" 可能带有戏谑或夸张的语气，反映出一种幽默或讽刺的情感(The expression "fully restrained" might carry a playful or exaggerated tone, reflecting a sense of humor or sarcasm)。

### 多模态线索(Multimodal Cues) ###
1. **结合视觉和语言信息(Combining Visual and Verbal Information)**:
  - 如果视频中的面部表情与"五花大绑"这一表达相结合，可能会增强对无助感的理解(If a facial expression in a video is paired with the phrase "tied up tightly," it may enhance the understanding of a sense of helplessness)。
  - 手势和姿态的变化（如表现出紧张或放松）与语言中的情感词汇相结合，可以更清晰地传达出说话者的情感状态，例如幽默、焦虑或讽刺(Changes in gestures and posture, such as showing tension or relaxation, combined with emotive language can more clearly convey the speaker's emotional state, such as humor, anxiety, or sarcasm)。

(b) Example 2

Figure 5: Examples of Prompts from MLLMs.

## D    LIMITATIONS

We found that training the PGMF-Student model is challenging. Achieving optimal performance requires careful tuning of hyperparameters. Additionally, as discussed in Section C.4, some loss values exhibit significant fluctuations during training with different random seeds, further highlighting the difficulty of training. This instability may also explain why the student models (*e.g.,* ALMT-Student in Section C.1) do not show significant improvements when PGMF is applied to other existing methods.

