# OpenReview forum: "Prompt-Guided Distillation from Multimodal Large Language Models to Task-specific Models for Multimodal Sentiment Analysis"
_ICLR.cc/2025/Conference — Submitted to ICLR 2025_

### Official Review · Reviewer_wMDu · 2024-11-04

**Soundness:** 3
**Presentation:** 3
**Contribution:** 3
**Rating:** 6
**Confidence:** 4

**Summary:**

The paper  propose a novel framework that integrates the generalized knowledge of MLLMs to guide smaller, task-specific models for better MSA.
﻿
Specifically, visual and audio features are aligned with language features via two alignment modules: Visual-to-Language Alignment and Audio-to-Language  Alignment. These conditional alignment layers establish correspondences between modalities with the help of the prompt, facilitating effective multimodal fusion with the help of MLLMs.
﻿
Both PGMF-Teacher and PGMF-Student can achieve good performance on two popular datasets (i.e., SIMS and MOSI), especially for PGMF-Student which can achieve improved performance without relying on prompt from MLLMs while maintaining fewer parameters.
﻿
This approach also offers a novel way to empower task-specific small models with the capabilities of MLLMs.

**Strengths:**

originality: The specific method of this paper has not been seen in other papers, so I believe it is original;

**Weaknesses:**

It is currently a common practice to distill knowledge from large models to small models and achieve improvement. This paper uses a multimodal large model to identify the clues that play a decisive role in predicting emotional labels in each modality, and then integrates them into the small model training framework for improvement. The idea is relatively straightforward, and the innovation is not particularly prominent.

**Questions:**

none

**Details Of Ethics Concerns:**

no concern

---

> ### Author Response · Authors · 2024-11-13
> **Response to Reviewer wMDu**
>
> ## **Response to Weakness**
>
> Thank you very much for your feedback for our work! We do appreciate your recognition of the originality of our work. Indeed, we are the first to introduce this specific framework that leverages MLLMs to help the training of the task-specific small-scale models for better MSA.
>
> **About the weakness, our choice of a simple and straightforward model structure was intentional.** By keeping the design clear and direct, we aimed to demonstrate the core effectiveness of our idea and framework itself. This can ensure that the improvements we observe are due to the framework **rather than any particular trick or complex structural design**. We believe that a simple yet effective method can also offer valuable contributions to the community, as has been shown by many solid works [3,4,5]. Our goal is to do a solid work which we hope can inspire further research and practical applications in MSA.

---

> ### Author Response · Authors · 2024-11-27
> **Follow-Up on Rebuttal of Reviewer wMDu**
>
> Dear Reviewer wMDu,
>
> We hope you are doing well. Thank you very much for your positive comments on our paper! We are writing to follow up on the response to our submission.
>
> We have submitted a revised version and responded to the reviewers' comments. **We would appreciate your kindly seeing our response and appreciate it if you would consider re-evaluating our paper.**
>
> If there are any remaining questions or if further clarification would be helpful, please do not hesitate to let us know.
>
> **We are looking forward to your response.**
>
> Best regards,
>
> The Authors

---

### Official Review · Reviewer_PNKv · 2024-11-05

**Soundness:** 3
**Presentation:** 2
**Contribution:** 3
**Rating:** 6
**Confidence:** 5

**Summary:**

The paper presents a distillation framework for Multimodal Sentiment Analysis, where knowledge is extracted from Multimodal LLMs and used to bias the attention maps of a smaller task-specific model. This model is then used as a teacher for a task-specific student model, which does not require the LLM-generated prompts during inference. The method achieves competitive performance on CMU-MOSI and state-of-the-art performance on CH-SIMS.

**Strengths:**

*  The presented method of prompt-guided attention transfer from MLLMs is a solid contribution and could be utilized for more tasks by the research community.
* Ablations regarding the effect of prompt-guidance, the effect of loss weight hyperparameters, and extension of the method to different architectures are included (though some are delegated to the appendix).
* Multiple-run averages, along with standard deviations are included.

**Weaknesses:**

* The evaluation could include the CMU-MOSEI dataset, which is larger and more recent than CMU-MOSI.
* The "Searched best seed" related rows in Tables 1 and 2 should be removed. Optimized seed results are rather uninformative, since the seed is not a tunable hyperparameter.
* I do not support the choice of delegating the "Related work" section to the Appendix, since relation to prior work should be a key component of a research paper. Reducing the size of figures 1, 2, removing  the optimized seed results, and performing a general revision of  the paper should create space in the main part of the paper.
* In general  Sections 2.4, 2.5 are a bit verbose and the equations are rather uninformative. Equation 11, especially so. What exactly is the fusion operation? I think these sections could benefit from a revision both in terms of concreteness and clarity and in terms of length.
* The  attention scores in Figure 3 range from -0.0004 to  0.0006 (very close to 0 and a difference from high to low score in the 4th decimal point). I think this is very concerning for the soundness of the method. What does the model actually attend to? Could this be due to the choice of the Hadamard product operation for fusing the attention matrices, which makes the scores extremely sparse / close to zero?

The authors have addressed most of these weaknesses in the rebuttal.

**Questions:**

My questions are included in the  "Weaknesses" section of the review

---

> ### Author Response · Authors · 2024-11-13
> **Response to Reviewer PNKv**
>
> ## **Response to W1**
> **Thank you for suggesting the inclusion of the CMU-MOSEI dataset for evaluation.** Due to the **financial cost** of using chatGPT API to help in training on such a large dataset, we faced some **economic constraints** during the initial experiments, which made it challenging to conduct evaluations on MOSEI. In response to your feedback, we have conducted experiments on the MOSEI dataset. **As shown in the table below (Table 3 of the paper),** the results on the larger dataset (MOSEI) show that PGMF-Teacher/-Student achieves advanced performance on most of the metrics with few parameters. This demonstrates that PGMF has good generalization ability on data sets of different sizes. It is worth noting that Self-MM with the fewest parameters shows well performance on the MOSEI dataset. This also demonstrates that the feasibility of suitable strategies to achieve strong performance with smaller parameters.
>
> | MOSEI|||||||
> | ------------ | --------- | ------------------------------ | ------------------------- | ------------------------- | -------------- | -------------- |
> | Method       | Parameter | Transformer-based Architecture | Acc-2| F1| MAE | Corr|
> | Video-LLaMA2 | 7B        | ✔️  | 83.29/84.50| 83.23/85.21  | 0.922| 0.406  |
> | GPT-4o-mini  | -         | ✔️ | **85.04/86.90** | **85.25/87.04**| **1.015**| **0.744**|
> | TFN          | 5.04M     | | 83.00±0.45/82.90±0.43| 82.68±0.40/82.83±0.41| 0.566±0.31| 0.725±0.21|
> | MISA         | 1.14M     | ✔️ | 84.41±0.30/85.09±0.62| 84.16±0.30/85.02±0.59 | 0.553±0.46     | 0.759±0.25|
> | Self-MM      | 0.16M     | | 84.15±0.50/84.90±0.49| 84.15±0.43/84.79±0.40     | **0.529±0.47** | 0.764±0.45|
> | TETFN        | 1.25M     | ✔️ | 84.18±0.62/85.42±0.43 | 84.06±0.63/85.31±0.55| 0.543±0.51| 0.769±0.27|
> | ALMT         | 3.21M     | ✔️ | 84.35±0.34/84.76±0.45| 84.10±0.32/84.25±0.59| 0.542±0.45| 0.768±0.17|
> | PGMF-Teacher | 1.47M     | ✔️ | **85.08±0.36/86.62±0.75** | **85.55±0.24/86.71±0.71** | 0.539±1.06| **0.773±1.51** |
> | PGMF-Student | 0.48M     | ✔️ | 83.96±0.38/84.67±0.27| 84.20±0.48/84.74±0.28| 0.548±0.41| 0.747±0.51|
>
> ## **Response to W2, W3 and W4**
> Thanks for your suggestion. We have made some effort to move the relevant work (Section 2) to the main body and ensure clarity and conciseness of the method presentation. The main revisions are: **1)** Remove the "Searched best seed" related rows and only report the average results of five runs. **2)** The size of Figures 1, 2, and 3 are reduced to make room for the related work. **3)** Remove some equations in Sections 3.4 and 3.5 to ensure clarity and conciseness. **4)** Explain explain the fusion operation in detail in Section 3.5. **The revised content can be found in the PDF of the revision rebuttal.**
>
> ## **Response to W5**
>
> 1. Figure 3 is an **attention difference map (not attention map)**, obtained by subtracting the attention map without MLLM guidance from that of the PGMF-Student. The values represent the difference in attention scores, with positive differences indicating areas where the PGMF-Student pay more attention to.
> 2. In the original attention map, each row’s attention scores sum to 1 after the softmax operation. However, due to the relatively long sequence length (e.g., 55 frames for video sequences in the SIMS dataset), the attention scores become more distributed across frames, resulting in a sparsity and low attention scores. Despite the low individual values, we can see from the difference map that these subtle changes in scores effectively shift the model’s focus across large regions, showing the prompt’s impact on attention distribution.
> 3. In long-sequence video data, we think changes between adjacent frames are continuous and gradual, making large attention score changes less possible. However, small changes in attention scores across many regions are sufficient to shift the model’s focus and significantly impact the multimodal alignment.
> 4. Our pipeline is intentionally designed to be simple, without any complex mechanisms in the alignment and fusion modules. This choice ensures that the improvements in performance are due to the framework itself, rather than relying on intricate structures or tricks. We believe this simplicity also demonstrates the effectiveness of our proposed framework.

---

> > ### Comment · Reviewer_PNKv · 2024-11-27
> > **Reply to authors**
> >
> > I thank the authors for addressing my comments. I am raising my score.

---

> > > ### Author Response · Authors · 2024-11-27
> > >
> > > Dear Reviewer PNKv,
> > >
> > > Thank you for adjusting your rating and supporting our work. We sincerely appreciate the opportunity to improve our submission and are grateful for the time and effort you have dedicated to reviewing it.
> > >
> > > Sincerely,
> > >
> > > The Authors

---

> ### Author Response · Authors · 2024-11-19
> **Updated Response and Rebuttal Revision for Manuscript**
>
> Hi, Reviewer PNKv. We have updated the responses, **including results on the MOSEI dataset and some revisions to the paper.**  The details can be found  in the PDF of rebuttal revision. **If you have any further questions, please do not hesitate to discuss them with us.** Thanks for your suggestion.

---

> ### Author Response · Authors · 2024-11-27
> **Follow-Up on Rebuttal of Reviewer PNKv**
>
> Dear Reviewer PNKv,
>
> We hope you are doing well. We are writing to follow up on the response to our submission. We appreciate your time and effort in reviewing our work. We know this is a busy time, but we would appreciate your kindly seeing our response. **We would appreciate it if you would consider re-evaluating our paper.**
>
> If there are any remaining questions or if further clarification would be helpful, please do not hesitate to let us know.
>
> **We are looking forward to your response.**
>
> Best regards,
>
> The Authors

---

### Official Review · Reviewer_9bUK · 2024-11-07

**Soundness:** 3
**Presentation:** 3
**Contribution:** 2
**Rating:** 5
**Confidence:** 4

**Summary:**

This study proposes a Prompt-Guided Multimodal Framework (PGMF) to transfer the capabilities of large Multimodal Large Language Models (MLLMs) to smaller, task-specific models for Multimodal Sentiment Analysis (MSA). PGMF consists of a teacher model (PGMF-Teacher) and a student model (PGMF-Student). The teacher uses MLLM-generated prompts to achieve better alignment and sentiment analysis, while the student learns to predict independently. Experiments show that PGMF-Teacher achieves state-of-the-art performance, while PGMF-Student achieves competitive results with fewer parameters, providing an efficient way to enhance small models with MLLM capabilities.

**Strengths:**

1. Leveraging Multimodal Large Language Models (MLLMs) to address the current challenges in the field of multimodal sentiment analysis represents a promising and worthwhile direction for exploration.
2. Embedding the teacher-student model paradigm within this domain is also a well-considered and potentially impactful approach.

**Weaknesses:**

1. It appears that your CONDITIONAL ALIGNMENT primarily facilitates attention-based interaction between the GPT-generated prompts and the corresponding content. However, I am unclear about the specific significance of taking the dot product of these two attention maps. From my perspective, your alignment module seems to merely apply attention mechanisms followed by a dot product, which does not appear to introduce any substantive algorithmic novelty. Could you elaborate further on the theoretical or empirical contributions this approach provides beyond the existing methods?

2. Your multimodal fusion module appears to simply concatenate features from different modalities and feed them into a transformer encoder. This approach is quite common and widely adopted in existing literature.

3. In a word, it seems that the paper primarily applies the teacher-student model paradigm to the domain of multimodal sentiment analysis (MSA), incorporating GPT-generated content as prompts. While the motivation is sound, the implementation appears somewhat simplistic, lacking sufficient innovation to substantiate a significant contribution.

4. The selection of baselines in your comparison is quite limited, and notably, none of the baselines are from 2024. Given that this field remains highly active and rapidly evolving, I strongly recommend including more recent baselines from 2024 to provide a more comprehensive and current evaluation of your proposed approach.

5. The analysis presented in the "EFFECT OF EACH COMPONENT" section appears rather superficial and lacks depth, raising the concern that it may have been generated by AI without sufficient refinement or critical examination.

6. Since the goal is to train a student model with reduced complexity, it would be highly informative to include a comparison of parameter counts with other baselines. Such a comparison would help substantiate claims regarding the efficiency and compactness of the student model relative to existing approaches.

**Questions:**

1. It appears that your CONDITIONAL ALIGNMENT primarily facilitates attention-based interaction between the GPT-generated prompts and the corresponding content. However, I am unclear about the specific significance of taking the dot product of these two attention maps. From my perspective, your alignment module seems to merely apply attention mechanisms followed by a dot product, which does not appear to introduce any substantive algorithmic novelty. Could you elaborate further on the theoretical or empirical contributions this approach provides beyond the existing methods?

2. Your multimodal fusion module appears to simply concatenate features from different modalities and feed them into a transformer encoder. This approach is quite common and widely adopted in existing literature.

3. In a word, it seems that the paper primarily applies the teacher-student model paradigm to the domain of multimodal sentiment analysis (MSA), incorporating GPT-generated content as prompts. While the motivation is sound, the implementation appears somewhat simplistic, lacking sufficient innovation to substantiate a significant contribution.

4. The selection of baselines in your comparison is quite limited, and notably, none of the baselines are from 2024. Given that this field remains highly active and rapidly evolving, I strongly recommend including more recent baselines from 2024 to provide a more comprehensive and current evaluation of your proposed approach.

5. The analysis presented in the "EFFECT OF EACH COMPONENT" section appears rather superficial and lacks depth, raising the concern that it may have been generated by AI without sufficient refinement or critical examination.

6. Since the goal is to train a student model with reduced complexity, it would be highly informative to include a comparison of parameter counts with other baselines. Such a comparison would help substantiate claims regarding the efficiency and compactness of the student model relative to existing approaches.

---

> ### Author Response · Authors · 2024-11-13
> **Response to Reviewer 9bUK**
>
> ## Response to Q1/W1
>
> **As shown in Figure 2 of the paper**, we first use the MLLMs' prompt as a query to extract essential information from other modalities, obtaining a shifted attention map △. This shifted attention map △ is then applied to the original attention map by the **dot product (it can be seen as an attention map fusion/transfer)**, effectively adjusting and optimizing the alignment process with the help of MLLMs. To verify the effectiveness of our idea, we choose the straightforward (dot product) to design the conditional alignment module. By employing a simple pipline, we focused on demonstrating that the MLLMs, even with basic structures/operations, could provide guidance that helps improve the task-specific small-scale model’s representation learning and overall performance. Specifically, the responses to contributions are as follows:
>
> 1. **As mentioned in General Response, the dot product within the conditional alignment is not the innovation point of the PGMF. Our focus is to validate the effectiveness of the framework through the straightforward design.** Instead, the core contribution lies in providing a novel way for MLLMs to directly participate in regulating the alignment process, helping the task-specific small-scale model focus on more relevant cross-modal relationships. This innovation combines MLLMs' prompts with cross-modal alignment, enabling more efficient completion of alignment tasks and improving performance in multimodal sentiment analysis. Although some existing methods [1,2] use MLLMs/LLMs to assist task-specific small-scale model training, they typically rely on MLLMs/LLMs to generate high-quality data. In contrast, our approach differs fundamentally in principle.
> 2. **As shown in Figure 3 of the paper**, we demonstrates the effect of the conditional alignment module by showing the difference between vision-language attention maps with and without the MLLMs' help. The difference map clearly indicates that the MLLMs can help the model focus more precisely on key regions in the language and visual modalities, demonstrating the effectiveness of this core idea. Additionally, experiments on the SIMS and MOSI datasets (**Table 1 and Table 2 of the paper, and Table 7 of the appendix**) show that the prompt-guided alignment module enables both the PGMF-Teacher and PGMF-Student to achieve state-of-the-art performance across most metrics. These results further confirm the empirical contribution of our framework.
>
> ## Response to Q2/W2
>
> **As mentioned in General Response**, our primary focus is not on the fusion method itself, but rather on how MLLMs are leveraged to help the learning of task-specific small-scale models. Therefore, **our innovation lies in the whole pipeline of the proposed framework PGMF.** Therefore, we opted for a simple way in the fusion module, using concatenation and a Transformer encoder, to emphasize the core impact of MLLMs in improving the task-specific small-scale model’s performance. **By achieving performance improvements** **without complex module design****, we believe the model's improvements in** **MSA** **are** **mainly derived from the framework**. We hope this clarification helps to convey the main contribution to our work.
>
> ## Response to Q3/W3
>
> **As mentioned in General Response**, **our core idea centers on the whole pipline** that using MLLMs to help the learning of task-specific small-scale models. **We intentionally chose the simplest method to validate our idea, which we believe can better demonstrate the effectiveness of our method.** By keeping the implementation straightforward, we can clearly show that the performance gains are due to the prompt-guided alignment **rather than any additional complex module designs**.
>
> In addition, Reviewer wMDu confirmed that our method has not been seen in other papers, it is original. Reviewer PNKv recognized that our method is with solid contribution and could be utilized for more tasks by the research community. These comments also demonstrate the novelty of our PGMF. **In a word, we believe that simple and effective methods are also valuable, simplicity is not an indication of lack of innovation. There are many works [3,4,5] that has simple architecture but make a great contribution to the community.**

---

> ### Author Response · Authors · 2024-11-13
> **Response to Reviewer 9bUK**
>
> ## Response to Q4/W4
>
> **In our initial submission, the most recent baseline with best results was ALMT.** It was published in December 2023 at EMNLP, which was **one of the best results with open-source code available** at the time. However, with recent releases from conferences like EMNLP 2024, we now have additional methods for comparison. Specifically, we have included the latest baselines for comparison, including KuDA [6] and FISFN [7]. **As shown in the table below**, we can see that PGMF-Teacher/Student achieves SOTA performance in all metrics on the SIMS dataset, demonstrating the effectiveness of our idea and framework. On MOSI datasets, the PGMF-Teacher/Student also can achieve good performance, especially in Acc-2 and F1. **In addition, it should be noticed that these recent methods are not open-sourced. So we are unable to conduct multiple runs to report the mean and standard deviation for a comprehensive comparison.**
>
> | SIMS         |                           |                           |                |                |
> | ------------ | ------------------------- | ------------------------- | -------------- | -------------- |
> | Method       | Acc-2                     | F1                        | MAE            | Corr           |
> | KuDA         | 80.74                     | 80.71                     | 0.408          | 0.613          |
> | FISFN        | 80.50                     | 80.7                      | 0.397          | 0.619          |
> | PGMF-Teacher | **83.06±0.95**            | **84.06±0.43**            | **0.370±0.50** | **0.690±0.80** |
> | PGMF-Student | 81.40±1.58                | 81.85±1.41                | 0.382±1.39     | 0.662±1.26     |
> | **MOSI**         |                           |                           |                |                |
> | KuDA         | 84.40/86.43               | 84.48/86.46               | **0.705**      | 0.795          |
> | FISFN        | 85.0/86.0                 | 85.0/86.0                 | 0.707          | **0.801**      |
> | PGMF-Teacher | **85.05±0.66/86.61±0.69** | **85.15±0.66/86.69±0.69** | 0.734±1.46     | 0.797±0.60     |
> | PGMF-Student | 83.62±0.91/85.37±1.00     | 83.68±0.96/85.50±0.96     | 0.746±1.63     | 0.775±1.10     |
>
> ## Response to Q5/W5
>
> **Thank you very much for your feedback on the "EFFECT OF EACH COMPONENT" section!** We would like to clarify that this paragraph was not generated by AI. In our submission, we condensed this section to meet length requirements, which may have inadvertently impacted the depth of the analysis. To address your concern, we have added detailed discussion in this section to provide a more critical analysis. In addition, we will reduce less essential content, such as removing single-run results and retaining only the multi-run averages and standard deviations for comparison, as suggested by Reviewer PNKv. The revised paragraph is as follows:
>
> > To evaluate the impact of each component within the framework, we conducted experiments by removing specific components. First, when we removed the MLLMs' prompt from the PGMF-Teacher, we observed a significant drop in performance across both datasets. Specifically, on the SIMS dataset, the F1 score decreased from 84.06% to 80.84%, and MAE increased from 0.370 to 0.436. A similar trend was observed on the MOSI dataset, where the F1 score dropped from 85.15% to 79.60%, and MAE increased from 0.734 to 0.914. These phenomenoa show that the MLLMs plays a crucial role in helping the model capture relevant multimodal information more effectively. Scond, we removed the guidance of the PGMF-Teacher during the training of the PGMF-Student. This led to a noticeable decrease in the student model's performance, with the F1 score on SIMS dropping from 81.85% to 78.72%, and on MOSI from 83.68% to 83.00%. The increase in MAE values on both datasets also reflects the PGMF-Student model's reduced ability to align multimodal information without teacher guidance. This result shows that the importance of knowledge distillation, as the PGMF-Teacher's guidance can help the PGMF-Student learn the relationship between each modality effectively.
> >
> > In addition, we also observed that the guidance from the PGMF-Teacher had a greater impact on the student model’s performance on the SIMS dataset compared to the MOSI dataset. We believe that this difference may be because of the diversity of data in the SIMS dataset. Specifically, the data of SIMS dataset contains complex environments and disturbances such as lighting, head pose and audio background noise. This makes the data difficult for the PGMF-Student to achieve better performance without relying on the guidance of the PGMF-Teacher.

---

> ### Author Response · Authors · 2024-11-13
> **Response to Reviewer 9bUK**
>
> ## Response to Q6/W6
>
> Thank you for highlighting the importance of comparing parameter counts with other baselines. **In Table 7 of the appendix**, we included a comparison with ALMT (the best-performing prior method). However, we also have realized that it is important to compare with more methods. Therefore, **as shown in Table blow**, we have added parameter count comparisons with other relevant methods. **In addition, it is worth noting that for the latest methods like** **CuDA** **[6] and FISFN [7] mentioned above, we were unable to include parameter counts due to the lack of open-source code.**
>
> As we can see from the table below, although PGMF-Student is with the second smallest parameter size only to Self-MM (a simple, direct and effective method), it achieves better performance. This demonstrates the effectiveness of our proposed framework and shows that PGMF achieves a balance of performance and parameters. In addition, we have also achieved significant improvements compared to the Transformer-based methods with few parameters.
>
> | SIMS         |           |                                |                           |                           |                |                |
> | ------------ | --------- | ------------------------------ | ------------------------- | ------------------------- | -------------- | -------------- |
> | Method       | Parameter | Transformer-based Architecture | Acc-2                     | F1                        | MAE            | Corr           |
> | TFN          | 35.63M    |                                | 78.12±1.56                | 77.83±1.62                | 0.434±1.12     | 0.579±1.50     |
> | MISA         | 21.66M    | ✔️                              | 77.72±1.10                | 76.54±1.67                | 0.451±1.83     | 0.570±1.95     |
> | Self-MM      | 0.38M     |                                | 77.94±1.11                | 77.72±0.68                | 0.418±1.05     | 0.589±1.54     |
> | TETFN        | 1.53M     | ✔️                              | 80.18±0.49                | 79.34±0.52                | 0.422±1.30     | 0.588±1.71     |
> | ALMT         | 2.60M     | ✔️                              | 79.91±0.29                | 80.17±0.60                | 0.421±0.69     | 0.583±0.70     |
> | PGMF-Teacher | 2.54M     | ✔️                              | **83.06±0.95**            | **84.06±0.43**            | **0.370±0.50** | **0.690±0.80** |
> | PGMF-Student | 0.82M     | ✔️                              | 81.40±1.58                | 81.85±1.41                | 0.382±1.39     | 0.662±1.26     |
> | **MOSI**         |           |                                |                           |                           |                |                |
> | Method       | Parameter | Transformer-based Architecture | Acc-2                     | F1                        | MAE            | Corr           |
> | TFN          | 9.50M     |                                | 77.38±1.37/78.11±0.60     | 77.35±1.33/78.02±0.57     | 0.949±3.13     | 0.662±1.95     |
> | MISA         | 1.14M     | ✔️                              | 80.93±0.99/81.05±0.83     | 80.90±1.03/81.01±0.87     | 0.773±1.81     | 0.775±0.63     |
> | Self-MM      | 0.16M     |                                | 82.94±0.63/83.18±0.35     | 82.95±0.63/83.09±0.36     | 0.717±1.53     | 0.792±0.55     |
> | TETFN        | 1.25M     | ✔️                              | 80.87±0.52/80.82±0.53     | 80.87±0.52/80.82±0.53     | 0.726±1.68     | 0.791±0.86     |
> | ALMT         | 2.50M     | ✔️                              | 83.00±0.22/85.12±0.20     | 83.00±0.22/85.19±0.27     | **0.713±0.75** | 0.795±0.54     |
> | PGMF-Teacher | 1.45M     | ✔️                              | **85.05±0.66/86.61±0.69** | **85.15±0.66/86.69±0.69** | 0.734±1.46     | **0.797±0.60** |
> | PGMF-Student | 0.53M     | ✔️                              | 83.62±0.91/85.37±1.00     | 83.68±0.96/85.50±0.96     | 0.746±1.63     | 0.775±1.10     |

---

### Author Response · Authors · 2024-11-13
**General Response**

# General Response

We appreciate the valuable time and effort from all reviewers, as well as the constructive comments and suggestions that contributed significantly to the improvement of our paper. **We are eager to engage in further discussions with you to address your concerns.**

**First, we would like to restate our motivation:**

We found that there are two limitations to apply MLLMs to MSA. 1) Although MLLMs have shown some improvement in MSA, their performance gains are often marginal and come at a high computational cost. 2) **Unlike the common practice in other fields** [1,2], using MLLMs to generate high-quality data for training small models is challenging for MSA due to the complexity of generating video, audio, and text data together. **These limitations** **motivate** **us to explore a different** **and more efficient** **direction: leveraging the guidance** **of** **MLLMs to help in training task-specific small-scale models for better MSA.** By involving MLLMs' prompt as guidance during the alignment and distillation process, we designed the PGMF that make the task-specific small-scale model to benefit from the MLLMs' knowledge. This presents a novel efficient framework.

**Second, we would like to restate the novelty is the framework rather than the model's architecture:**

1. At the framework level, our pipeline **for the first time** utilizes prompt outputs from MLLMs as conditional guidance within a teacher-student framework, effectively improving the alignment and learning process of the student model. The experimental results validate the effectiveness of this framework, demonstrating that it enables the student model to achieve state-of-the-art performance with reduced complexity.
2. **We intentionally chose the simple** **and straightforward** **architecture modules** **to validate our framework**,which we believe can better demonstrate the effectiveness of our framework. By keeping the implementation straightforward, we can clearly show that the performance gains are due to the prompt-guided alignment **rather than any additional complex module designs**.
3. **We believe that simplicity and effectiveness are valuable for the community.** There are many works [3,4,5] that have simple architecture but make a great contribution to the community. For example, in the MSA field, Self-MM [5] achieves significant performance despite using only MLPs in its network architecture. Similarly, our framework PGMF, although intentionally designed to be simple, offers a new perspective on using MLLMs to guide smaller models in multimodal tasks, which we believe can inspire further exploration and refinement in this area.

Thank you for your patience and suggestions. **We look forward to discussing our work with you.**

Sincerely,

The Authors


# Reference Used throughout the Rebuttal

[1] Chen, L. et al., 2023. Sharegpt4v: Improving Large Multi-modal Models with Better Captions. *arXiv* *preprint**arXiv:2311.12793*.

[2] Chen, L. et al., 2024. Sharegpt4video: Improving Video Understanding and Generation with Better Captions. *arXiv* *preprint* *arXiv:2406.04325*.

[3] Zadeh A. et al., 2017. Tensor Fusion Network for Multimodal Sentiment Analysis. In EMNLP 2017.

[4] Yao-Hung Hubert Tsaiet al., 2019. Multimodal Transformer for Unaligned Multimodal Language Sequences. In ACL 2019.

[5] Wenmeng Yu et al., 2021. Learning Modality-specific Representations with Self-supervised Multi-task Learning for Multimodal Sentiment Analysis. In AAAI 2021.

[6] Feng, X. et al., 2024. Knowledge-Guided Dynamic Modality Attention Fusion Framework for Multimodal Sentiment Analysis. In EMNLP 2024.

[7] Jin, Y., 2024. GSIFN: A Graph-Structured and Interlaced-Masked Multimodal Transformer Based Fusion Network for Multimodal Sentiment Analysis. *arXiv* *preprint* *arXiv:2408.14809*.

---

### Author Response · Authors · 2024-11-23
**Follow-up on Review Feedback**

Dear Reviewers,

I hope this message finds you well. We sincerely appreciate the time and effort you have dedicated to reviewing my work. As the rebuttal deadline approaches, **we would greatly appreciate it if you could kindly provide feedback on our responses to your initial comments**. Your insights would be invaluable in helping us refine and strengthen our work.

Thank you very much for your time.

Best regards,

The Authors

---

> ### Comment · Reviewer_9bUK · 2024-11-25
>
> Thank you for your detailed response and the additional experiments provided. I have carefully reviewed your rebuttal, and I appreciate the effort and thoughtfulness that went into addressing the initial comments.
>
> Your approach to leveraging LLM to train a lightweight model for multimodal sentiment analysis is indeed meaningful and has potential. However, considering the current version of the work, I believe it aligns more with a score of 5 at best. ICLR places significant emphasis on algorithmic innovation, and I feel that aspect could be further strengthened in your submission.
>
> Thank you again for your efforts and engagement.

---

> > ### Author Response · Authors · 2024-11-25
> > **Response to Reviewer 9bUK**
> >
> > Thank you for taking the time to review our rebuttal and for your feedback.
> >
> > We respect your perspective on algorithmic innovation being a core criterion for ICLR submissions. **However, we also believe that impactful contributions of a paper have diverse forms, including framework development, benchmarking, and empirical insights.** Our proposed PGMF first integrate MLLMs with task-specific models, providing a novel and efficient way for better MSA.
> >
> > Finally, we do sincerely thanks for your time, feedback and the opportunity to discuss our work.
> >
> > Sincerely,
> >
> > The Authors

---

### Meta-Review · Area_Chair_jw5b · 2024-12-24

**Metareview:**

The paper proposes a distillation approach for Multimodal Sentiment Analysis, where knowledge is extracted from Multimodal LLMs and used to bias the attention maps of a smaller task-specific model.

Two reviewers vote to borderline acceptance and one reviewer gave borderline rejection while pointing out some limitations such as limited originalty and lack of innovation.

Even if this paper in on the borderline, considering the ICLR quality, the contributions seem not to be sufficient for ICLR presentation.

So, AC recommends rejecting this paper.

**Additional Comments On Reviewer Discussion:**

Initial scores were 6, 5, and 3.

Main concerns raised by two reviewers were lack of experimental results, paper organization, unclear method description and analisys,  lack of innovation in methodology.

During the rebuttal, the scores became 6, 6, and 5.
Via AC-reviewer discussion, 9bUK argued that the contribution of this paper is not sufficient for ICLR quality with score 5.

---

### Decision · Program_Chairs · 2025-01-22

Reject